# Real-Time Novel-View Freehand Ultrasound Imaging via Point-Cloud Rendering and Diffusion-Bridge Completion

**Hanrui Shi**[1]                                                                       HANRUISH@UW.EDU
*University of Washington, Electrical & Computer Engineering, Seattle, WA 98105, USA*

**Boris Mailhé**                                                                     BORIS.MAILHE@UII-AI.COM
**Zheyuan Zhang**                                                           ZHEYUAN.ZHANG01@UII-AI.COM
**Yikang Liu**                                                                        YIKANG.LIU@UII-AI.COM
**Xiao Chen**                                                                       XIAO.CHEN01@UII-AI.COM
**Ankush Mukherjee**                                                     ANKUSH.MUKHERJEE@UII-AI.COM
**Terrence Chen**                                                               TERRENCE.CHEN@UII-AI.COM
**Shanhui Sun**                                                                   SHANHUI.SUN@UII-AI.COM
*United Imaging Intelligence, Boston, MA 01803, USA*

**Editors:** Under Review for MIDL 2026

## Abstract

Freehand ultrasound imaging is limited by sparse sampling and restricted probe coverage, which prevent consistent visualization of unseen planes and oblique orientations. We propose a real-time framework for novel-view ultrasound imaging that combines point-cloud rendering with diffusion-bridge completion. Given a sequence of 2D B-mode images and tracked probe poses, each novel view is first rendered as a partially observed slice from the reconstructed point cloud geometry, then completed by an Image-to-Image Schrödinger Bridge ($I^2SB$) model to synthesize anatomically coherent textures. The diffusion-bridge formulation accelerates convergence by conditioning on visible regions instead of noise, enabling stochastic yet efficient generation. A latent $I^2SB$ variant further improves computational efficiency for high-resolution ultrasound data. Experiments on an abdominal dataset demonstrate realistic novel-view synthesis with fine structural continuity and real-time inference ($<0.2$ seconds per view), outperforming standard diffusion inpainting baselines in both speed and visual fidelity. The proposed method provides an efficient generative approach for interactive and view-adaptive ultrasound visualization.

**Keywords:** ultrasound imaging, novel-view synthesis, diffusion bridge, point-cloud reconstruction, real-time imaging

## 1. Introduction

Ultrasound imaging is widely used in clinical diagnostics and interventional procedures due to its real-time capability, portability and safety. However, freehand acquisition only samples one slice at a time as shown in Figure 1, and the sonographer must navigate to the target region of interest based on this limited information. Multiple navigation assistance methods have been proposed, from navigation guidance (Droste et al., 2020) to autonomous robotic scanning (Jiang et al., 2023). To be most useful, such methods must function for arbitrary views, not just good quality ones acquired during a typical clinical scan. Therefore,

---

1. This work was carried out during the internship of the author at United Imaging Intelligence, Boston, MA

SHI[1] MAILHÉ ZHANG LIU CHEN MUKHERJEE CHEN SUN

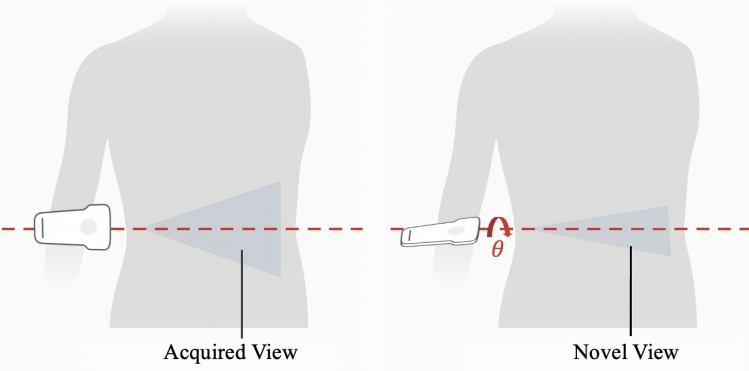

Figure 1: Illustration of an acquired view and an example novel view. The acquired view reflects the probe's true imaging plane, whereas the novel view is synthesized by rotating the probe orientation by an angle $\theta$.

enabling novel-view ultrasound imaging from limited freehand inputs is an important step for data augmentation and for improving the performance and robustness of ultrasound navigation systems.

Task-specific GAN-based synthesis methods have been proposed for training data generation / augmentation conditioned on segmentation masks obtained from a small number of ultrasound images (Tiago et al., 2022) or from an external CT or MRI dataset (Amadou et al., 2024; Peng et al., 2019). Image inpainting has also been used to edit annotations out of secondary capture ultrasound images (Chen et al., 2023). Conventional approaches for deriving unseen ultrasound views rely on interpolation or grid-based resampling, yet these methods struggle to reconstruct realistic content in regions that were never directly sampled, leading to blurred or incomplete appearances (Solberg et al., 2007; Cong et al., 2017; Prager et al., 1999). Kernel-regression techniques provide smoother estimates but remain highly dependent on dense and uniform sampling(Chen et al., 2014; Wen et al., 2013, 2018). GAN view generation has also been proposed for 2D fetal scans with optical tracking (Hu et al., 2017), however, the proposed conditioning on view poses rather than image content is unlikely to enable generalization to new poses or subjects. More recently, implicit-field and learning-based representations such as NeRF and Gaussian splatting (Wysocki et al., 2024; Dou et al., 2025; Yeung et al., 2024; Eid et al., 2025) have shown potential for novel-view synthesis by modeling ultrasound appearance continuously across viewpoints. However, these methods typically require per-volume optimization and are therefore impractical for real-time inference of 2D slices. Moreover, because they rely on consistency with acquired rays, they cannot reliably infer structures in regions that were not acoustically sampled. Generative diffusion models offer stronger priors for synthesizing realistic textures(Ho et al., 2020; Lugmayr et al., 2022; Saharia et al., 2022), but their iterative sampling remains computationally expensive. Contemporary to our work, diffusion-based inpainting has also been proposed to accelerate 3D ultrasound scanning(Stevens et al., 2025). While both inpainting methods bear some similarity, the target applications are quite different since even accelerated 3D scans are restricted to the field of view of the 3D probe. Diffusion-bridge

methods(Chen et al., 2021; Liu et al., 2023) provide a more efficient alternative by conditioning on partially observed inputs, yet have not been explored for real-time ultrasound novel-view synthesis.

To address these limitations, we propose a real-time novel-view ultrasound imaging framework that integrates point-cloud rendering with diffusion-bridge completion. Acquired B-mode pixels are first projected into 3D space to form a dense point cloud, from which novel viewpoints are rendered as partially observed slices using forward splatting. An Image-to-Image Schrödinger Bridge ($I^2SB$) model is then trained with the fully acquired ultrasound slices to stochastically complete these slices, generating anatomically consistent textures beyond the observed regions. A latent $I^2SB$ variant is further proposed to accelerate processing for high-resolution ultrasound images. We demonstrate the effectiveness of the proposed pipeline and show that the diffusion-bridge formulation achieves superior reconstruction quality and substantially faster inference compared with standard diffusion-based completion. Together, these components enable efficient, view-adaptive ultrasound visualization and data augmentation from sparse freehand acquisitions. Our main contributions are summarized as follows:

- A real-time synthesis pipeline for novel-views from freehand ultrasound, enabled by coupling geometry-aware point-cloud rendering with a diffusion-bridge completion mechanism that produces anatomically consistent slices at <0.2 seconds per view.

- A high-resolution latent diffusion-bridge model that performs completion in a compact latent space, preserving fine anatomical details while greatly reducing computation cost and enabling full-resolution real-time inference.

- A comprehensive experimental study on a large abdominal ultrasound dataset, including quantitative benchmarks, uncertainty analysis, and efficiency evaluations, demonstrating substantial gains in both fidelity and speed over diffusion-based inpainting baselines.

## 2. Methods

### 2.1. Novel View Synthesis from Point Cloud Rendering

As shown in Figure 2, starting from the sequence of images acquired during an ultrasound sweep and the 6 degrees of freedom pose of the ultrasound probe at every time point, novel views are generated by extracting slices from a reconstructed 3D volume.

To reconstruct the volume, the world coordinates of all non-zero pixels of every image are first computed using a calibrated intrinsic matrix $K = \begin{bmatrix} \Delta & c \\ 0 & 1 \end{bmatrix}$ and tracked probe pose $T = \begin{bmatrix} R & t \\ 0 & 1 \end{bmatrix}$. The intrinsic matrix $K$ converts pixel indices to coordinates in a probe coordinate system, with $\Delta$ the pixel spacing of the probe and $c$ the offset between the image corner and the attachment position of the tracker on the probe. The through-plane pixel index is 0 for the 2D ultrasound probes considered in this work. Each pixel defines a point with image-plane coordinates $p$, which is mapped to a spatial position $p_w = TKp$, and

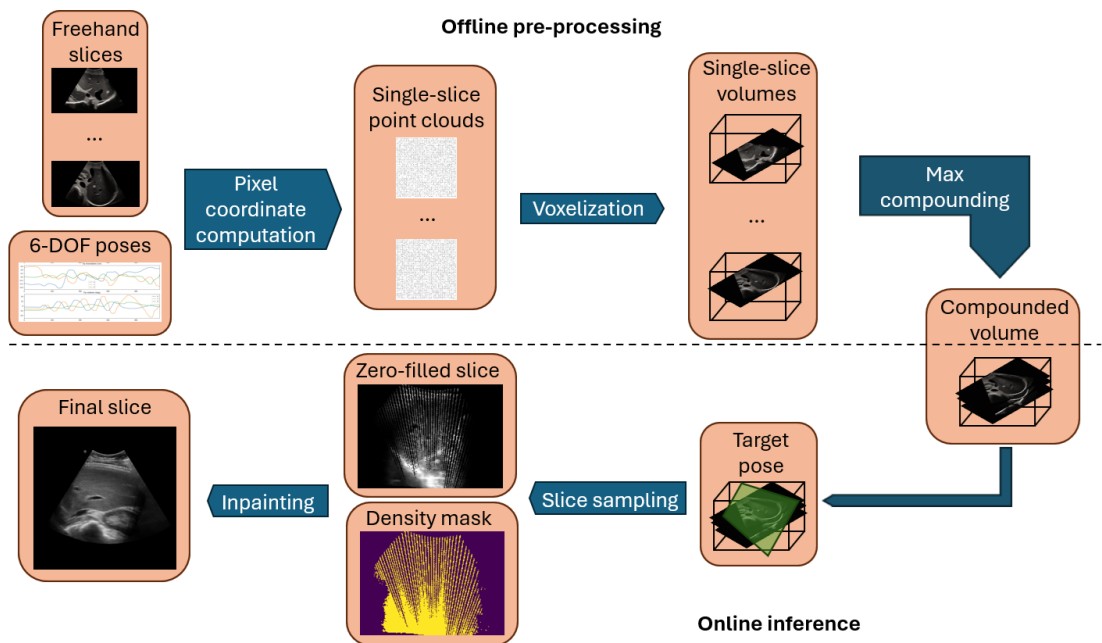

Figure 2: Overview of the Proposed Freehand Ultrasound Novel-View Generation Pipeline. Multiple freehand ultrasound slices are projected into point clouds, gridded and compounded into a 3D volume, and re-sliced to produce partial novel views and their masks. These incomplete slices are refined by an $I^2SB$ completion model, which is trained using simulated partial views derived from acquired slices and randomly sampled novel-view masks.

carries an associated ultrasound intensity $f(p)$. All the points from acquired slices form the point set $\mathcal{P}$.

A common voxel grid for all images is computed using the bounding box of all pixel coordinates and a fixed voxel spacing. Images are resampled onto that grid, producing a volume $F \in \mathbb{R}^{I_x \times I_y \times I_z}$ . We implemented this resampling using PyTorch3D[1]: each image is first converted to a point cloud of pixels then voxelized. Voxelization also generates a density mask $D$ over the grid that counts the number of points contributing to a voxel, weighted by the trilinear resampling kernel. In order to provide robustness to shading artifacts from ultrasound acquisition, the volumes from each image are combined using max compounding, which assigns each voxel the maximum intensity among its contributing samples:

$$F(i_x, i_y, i_z) = \max_{p \in \mathcal{P}_{i_x, i_y, i_z}} f(p)$$

$$D(i_x, i_y, i_z) = \max_{p \in \mathcal{P}_{i_x, i_y, i_z}} d(p)$$

---

1. https://pytorch3d.org/

with $f$ the voxel intensities and $d$ the voxel densities. This preserves strong echo boundaries, naturally handles occlusions, and reduces sensitivity to probe pressure or gain variations when multiple frames contribute to the same voxel.

Novel views at arbitrary poses are generated by computing the corresponding pixel world coordinates and interpolating their intensities and densities from the 3D volume. A binary visibility mask is then constructed from the point-cloud density map, where voxels with normalized density below a threshold are labeled as unobserved and marked as missing in the rendered slice. The threshold is determined from the bimodal density histogram computed on the acquired in-plane slices, with the valley between the signal and background modes selected as the decision boundary. The resulting partially observed slice, together with its mask, is subsequently fed into the diffusion-bridge completion model.

## 2.2. I$^2$SB for Novel-View Completion

To address the incomplete spatial coverage present in novel-view slices, we employ an I$^2$SB formulation for stochastic inpainting. The model operates directly on the partially observed slices produced by point-cloud rendering and aims to synthesize anatomically plausible completions that are consistent with both the visible pixels and the learned distribution of fully acquired ultrasound images.

**Problem Formulation:** Let $\mathbf{x}_0$ denote a simulated slice with missing regions encoded by a visibility mask $\mathbf{m}$, and let $\pi_1(\mathbf{x}_1)$ represent the distribution of fully sampled in-plane ultrasound slices. The completion task is framed as learning a continuous stochastic process that transports from the conditional distribution $\pi_0(\mathbf{x}_0|\mathbf{m})$ to $\pi_1(\mathbf{x}_1)$. Specifically, I$^2$SB learns a generative bridge $\{p_t(\mathbf{x}_t)\}_{t=0}^1$ that interpolates between these distributions through a sequence of forward and reverse diffusion. The forward process gradually perturbs the masked slice toward a reference distribution, while the reverse process reconstructs missing structures by conditioning explicitly on the mask, thereby ensuring that synthesis remains compatible with the observed pixels.

**Model Design:** Following the diffusion-bridge framework, I$^2$SB differs from standard denoising diffusion models in that sampling does not begin from pure Gaussian noise. Instead, the reverse trajectory is initialized from the partially observed input, allowing the model to focus its stochastic updates on the missing regions. During training, visibility masks $\mathbf{m}$ are sampled from out-of-plane slices reconstructed by the rendering pipeline; these masks reflect realistic spatial patterns of missing data that occur during freehand scanning. At each diffusion step $t$, the network predicts the score function needed to reverse the diffusion process, and is optimized using the Schrödinger bridge objective:

$$\mathcal{L}_{\mathrm{SB}} = \mathbb{E}_{t,\mathbf{x}_t} \left[ \|s_\theta(\mathbf{x}_t, t, \mathbf{m}) - \nabla_{\mathbf{x}_t} \log p_t(\mathbf{x}_t|\mathbf{m})\|_2^2 \right].$$

At inference time, the learned bridge performs stochastic sampling that progressively refines the partially observed slice. Because the model conditions on the visible regions throughout the denoising trajectory, it naturally preserves geometric features already present in the data while synthesizing plausible textures in the unobserved areas.

**Latent I$^2$SB:** To further accelerate inference for high-resolution slices, we introduce a latent diffusion-bridge variant in Figure 3. In this formulation, the input slice is first mapped into a compact latent representation using a pretrained VAE encoder, after which

the I$^2$SB model performs completion entirely within this lower-dimensional space. The decoder then reconstructs the completed slice back to image space. This approach substantially reduces computational load while maintaining structural fidelity, as the latent representation captures the essential anatomical content of the ultrasound images. Operating in the latent domain also enables real-time performance for full-resolution synthesis, making the approach well suited for interactive imaging applications.

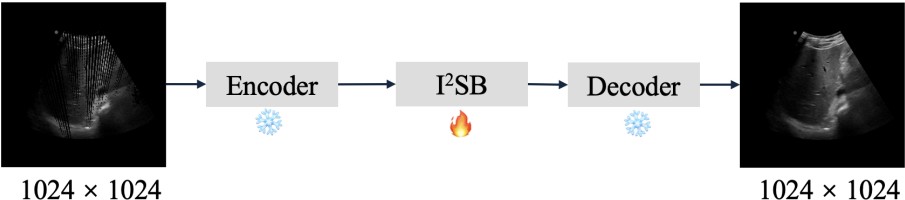

1024 × 1024           1024 × 1024

Figure 3: Framework of the latent I$^2$SB model. A pretrained VAE encoder–decoder pair is employed to process the inputs and outputs of I$^2$SB, reducing computational overhead.

## 3. Experiments and Results

### 3.1. Experimental Setup

**Dataset.** Experiments were conducted on a private abdominal ultrasound dataset containing multi-sweep freehand scans from 175 subjects, each acquired under routine clinical protocols. For every sweep, the 6-degree-of-freedom probe poses were recorded using an electromagnetic tracking system, enabling accurate mapping between 2D pixel coordinates and 3D spatial locations. Each B-mode slice provides an in-plane resolution of $0.25 \times 0.25$ mm and a raw size of $900 \times 1100$ pixels, capturing fine-scale abdominal structures. To balance fidelity and computational cost, the vanilla I$^2$SB model operated on slices downsampled to $1 \times 1$ mm resolution, while the latent I$^2$SB variant processed full-resolution inputs through the VAE encoder. In total, approximately 400k acquired slices were included in the training set. For each slice, a synthetic partial observation mask was generated by the novel-view rendering pipeline, emulating the visibility patterns typically encountered in sparse freehand acquisition. The dataset was partitioned into training, validation, and test sets using an 8:1:1 ratio.

**Implementation.** All experiments were performed on a single NVIDIA V100 GPU (16 GB). Point-cloud rendering and voxelization were implemented using the PyTorch3D framework, which provides efficient GPU-accelerated routines for forward splatting, density accumulation, and trilinear interpolation.Volume generation is a one-time cost, requiring approximately 1 ms per slice (e.g., 0.5 s to fuse 500 slices into a single volume). Reslicing from the volume completes in under 10 ms.

The completion model follows the I$^2$SB formulation, with a U-Net backbone that estimates diffusion scores in both pixel and latent domains. For the latent I$^2$SB variant, we directly adopt the pretrained VAE[2] used in Stable Diffusion (Rombach et al., 2022) without

---

2. https://huggingface.co/stabilityai/sd-vae-ft-mse

further finetuning, to map images into a compact latent space and reconstruct completed slices, which yields stable color fidelity with minimal reconstruction bias. Models were trained for approximately 48 hours using a batch size of 4 and the Adam optimizer with a learning rate of $1 \times 10^{-4}$, together with standard data normalization and random mask perturbations to improve robustness. Inference time scales linearly with the number of reverse diffusion steps, requiring roughly $0.03 \times r$ seconds per synthesized view (batch size = 1), and fits comfortably within 4 GB of GPU memory. With as few as $r = 5$ steps, the model produces visually stable and anatomically coherent inpainting, enabling near real-time performance suitable for interactive applications.

**Evaluation.** To evaluate the proposed completion framework,we compare I$^2$SB with conventional linear interpolation as well as two diffusion-based inpainting methods: RePaint and Palette. RePaint performs unconditional diffusion with ground-truth value replacement and resampling during inference, while Palette conditions on the incomplete image for noise prediction. We include these two methods as representative diffusion-based inpainting baselines. For supervised in-plane completion, where paired ground-truth slices are available, we measure reconstruction fidelity using SSIM, PSNR and LPIPS. For novel-view completion, in which no explicit ground truth exists, we generate 4k synthetic views and assess perceptual consistency with the acquired slices using FID. In addition, we estimate epistemic uncertainty by performing Monte Carlo sampling with repeated stochastic inference (1000 realizations per sample for each generative method) and computing pixel-wise variance maps to characterize the stability of the generated completions under different noise realizations.

## 3.2. Results

We present synthesized novel-view results obtained with different completion methods and analyze their performance across resolution settings, perceptual quality, and computational efficiency. Linear interpolation, RePaint, Palette and the vanilla I$^2$SB model are evaluated in a downsampled configuration to fit the available computational budget, whereas the latent I$^2$SB variant is assessed at full spatial resolution by operating directly within a compressed latent space. In addition to comparing the final synthesized images, we also examine how key hyperparameters influence the trade-off between reconstruction fidelity and runtime, which is crucial for real-time ultrasound applications.

**Qualitative Results.** Figure 4 presents representative synthesized novel-view slices produced by the different completion models. The partially observed slices generated through point-cloud forward splatting exhibit complex and irregular missing regions, directly reflecting the heterogeneous spatial coverage typical of freehand scanning. These incomplete views therefore constitute a challenging testbed for evaluating each model's ability to infer anatomically plausible structures without direct observations.

All methods visually succeed in filling the missing areas at a global level because the regions to be completed are relatively small. However, closer inspection reveals that linear interpolation smooths away fine details. RePaint and Palette produce globally coherent textures but tend to oversmooth the fine-scale structures characteristic of abdominal ultrasound, resulting in blurred organ boundaries and reduced visibility of subtle anatomical features. Palette offers better results than RePaint. In contrast, the pixel-space I$^2$SB model

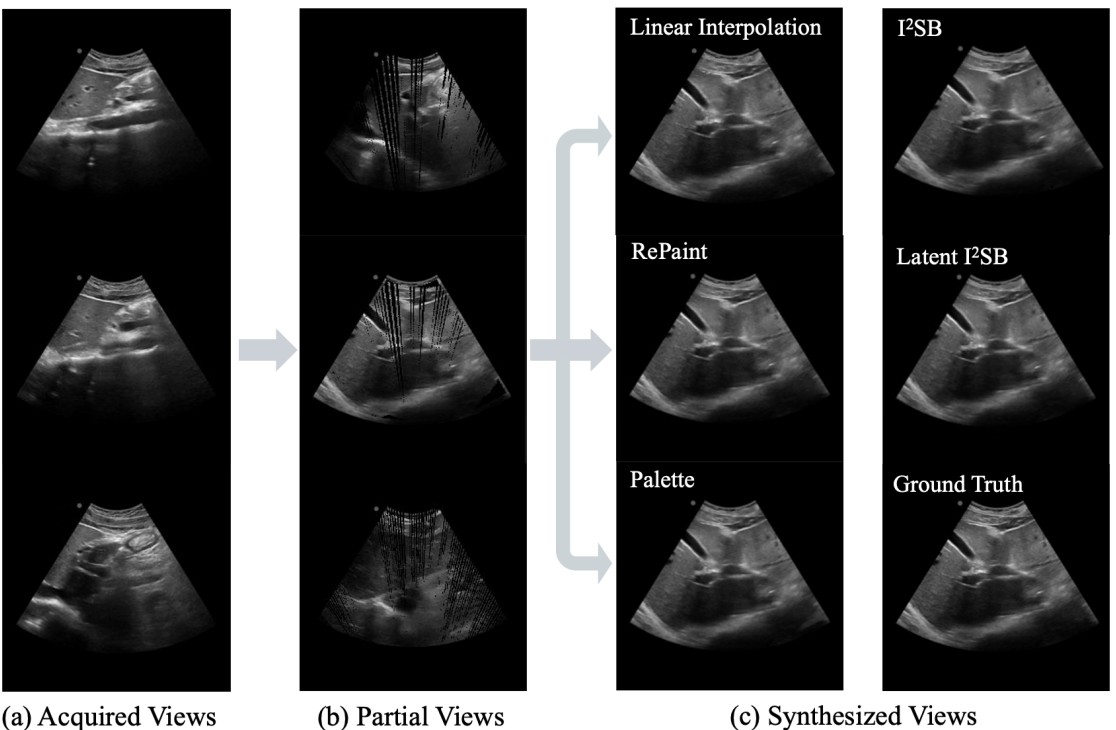

Figure 4: Novel-view ultrasound synthesis results. (a) Acquired views from sparse freehand scanning. (b) Random partial views rendered from the point-cloud representation, showing missing regions prior to completion. (c) Synthesized views produced by different methods, compared with the ground truth.

reconstructs sharper interfaces and more faithfully reproduces speckle statistics, yielding completions that are visually closer to the acquired slices. The latent $I^2SB$ variant achieves comparably detailed reconstructions while operating in a compressed latent domain, demonstrating that high-level geometric and anatomical cues can be effectively preserved even under reduced computational overhead. These results highlight the advantage of diffusion-bridge conditioning, which provides stronger and more stable guidance from observed pixels into missing regions, thereby promoting anatomically continuous and spatially coherent reconstructions. Figure 5 further illustrates this behavior by showing the bridge trajectory, where the intermediate states progressively recover the structure and converge towards a realistic finished slice.

**Quantitative Comparison.** Table 3.2 reports quantitative performance across SSIM, PSNR, LPIPS, and FID. The vanilla $I^2SB$ model achieves the highest fidelity across all metrics, indicating that the bridge formulation enables more accurate restoration of both global structure and fine-grained texture. Linear interpolation obtains high SSIM and PSNR because these metrics reward smooth and pixel-aligned outputs. Nevertheless, perceptual metrics such as LPIPS and FID reveal a non-negligible discrepancy between diffusion-based reconstructions and the ground truth in terms of perceptual fidelity. RePaint shows mod-

$x_0$ $\longrightarrow$ $x_1$

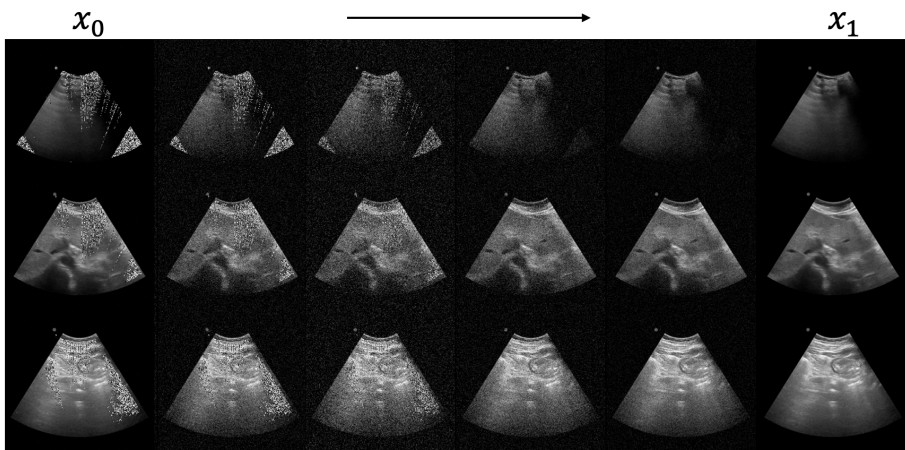

Figure 5: I$^2$SB Trajectory for Novel View Completion. The stochastic bridge learned to transition from the masked novel view $x_0$ to the complete view $x_1$. Early states exhibit noise and uncertainty in missing regions, while later states progressively recover consistent structures and realistic speckle patterns.

Table 1: Quantitative comparison of completion methods. The best performance is highlighted in **bold**.

| Method | Supervised | | | Unsupervised |
|---|---|---|---|---|
| | SSIM $\uparrow$ | PSNR $\uparrow$ | LPIPS $\downarrow$ | FID $\downarrow$ |
| w/o Completion | 0.6052 | 11.38 | 0.5578 | 155.80 |
| Linear Interpolation | 0.9422 | 34.20 | 0.0482 | 24.05 |
| RePaint | 0.8998 | 29.15 | 0.0688 | 64.57 |
| Palette | 0.9676 | 34.28 | 0.0467 | 19.44 |
| I$^2$SB | **0.9798** | **39.15** | **0.0172** | **2.113** |
| latent I$^2$SB | 0.9456 | 37.25 | 0.0312 | 32.19 |

erate improvements over the uncompleted slices but remains substantially limited in perceptual metrics, consistent with its tendency to over-smooth textures. Palette is better than RePaint due to the benefits of conditional training. The latent I$^2$SB variant, despite operating at full resolution, attains competitive performance while using far fewer computational resources. This suggests that latent-space modeling is well suited for ultrasound completion tasks, where structural information dominates and high-frequency content can be effectively reconstructed through decoding.

**Uncertainty Analysis.** Figure 6 shows the pixel-wise variance maps of a representative sample of Monte Carlo sampling. RePaint and Palette exhibit elevated variance, particularly

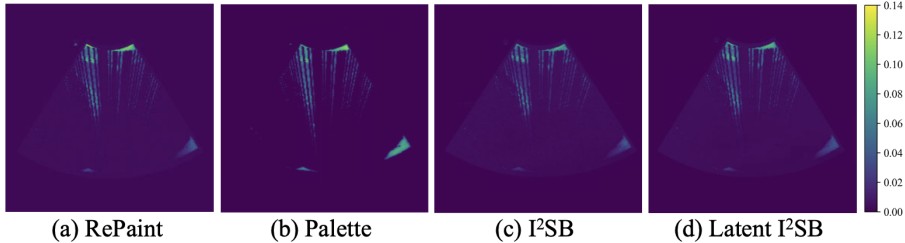

Figure 6: Variance maps from Monte Carlo sampling. I$^2$SB models exhibit higher sample consistency than RePaint and Palette.

along tissue boundaries, shadowed regions, and occluded zones. This indicates that their completion behavior is more sensitive to sampling noise and less constrained by the observed input.

Both pixel-space and latent I$^2$SB models produce significantly lower and more localized variance, reflecting tighter adherence to the observed pixels and more stable conditional sampling trajectories. Notably, the latent I$^2$SB variant preserves the uncertainty profile of the full-resolution model despite performing denoising in a compressed space, underscoring the robustness of the bridge formulation under dimensionality reduction. This stability is especially valuable for clinical or real-time settings where consistent predictions across repeated inference runs are essential.

**Step Reduction for Real-Time Completion.** Linear interpolation does not involve model inference and is therefore the fastest method, but it inevitably loses realistic fine-scale details. Palette delivers strong perceptual performance, but a full inference pass typically takes about 25 seconds. For this reason, we conduct a trade-off analysis between inference time and reconstruction quality for RePaint and I$^2$SB. Figure 7 illustrates these relationships. For I$^2$SB, SSIM increases rapidly as the number of replaced diffusion steps increases, with performance saturating around five steps, which yields an average inference time of 0.15 seconds per image, enabling real-time completion. This plateau suggests that the bridge formulation effectively minimizes the number of required sampling iterations, making real-time deployment feasible.

By contrast, RePaint requires substantially more steps and repeated resampling to approach similar performance, leading to runtimes that exceed several seconds per slice. Although the method eventually stabilizes, its computational overhead makes it unsuitable for interactive applications. To ensure equitable comparisons across models, we adopt a matched inference-time configuration with five diffusion steps and a resampling rate of 1 for RePaint in all quantitative experiments.

## 4. Discussion and Conclusion

Our results demonstrate the effectiveness of novel-view ultrasound synthesis under sparse freehand acquisitions. By combining point-cloud rendering with diffusion-bridge completion, the proposed framework preserves geometric consistency while probabilistically filling in missing local details. This hybrid formulation produces structurally coherent and

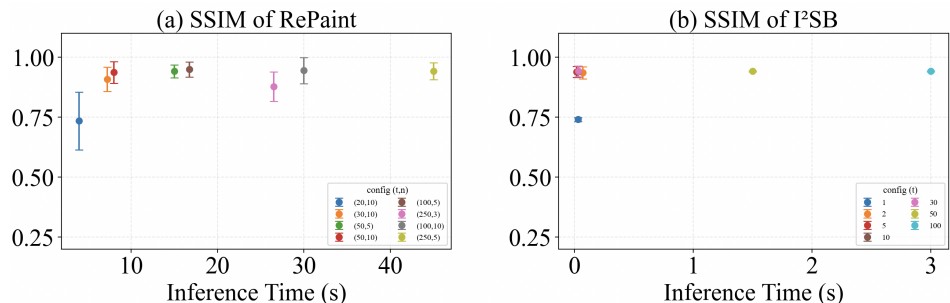

Figure 7: Comparison of SSIM versus inference time between RePaint (a) and I$^2$SB (b). The proposed I$^2$SB achieves comparable SSIM with substantially reduced inference time.

anatomically aligned views, indicating that the pipeline can robustly recover high-quality novel perspectives even from limited and irregular freehand sampling.

Beyond validating the pipeline, our experiments highlight the advantages of diffusion-bridge–based completion. I$^2$SB achieves the highest quantitative performance and lower uncertainty compared with traditional diffusion inpainting, indicating that the bridge constraint produces more stable and anatomically consistent completions. Its latent-space variant extends these benefits to full-resolution inference with substantially reduced computational cost. Although the encoder–decoder structure introduces minor and unavoidable information loss, the overall completion quality remains strong. Importantly, step-reduction analysis shows that I$^2$SB reaches performance saturation with as few as five diffusion steps, allowing real-time completion, which is critical for interactive ultrasound applications such as probe guidance or rapid 3D volume reconstruction.

Despite these strengths, there remain limitations. First, diffusion-based completion may produce diverse yet equally plausible outputs when large spatial gaps are present, reflecting inherent ambiguity rather than model error. This behavior is expected, but it implies that completion quality depends on having at least moderate geometric support from the point-cloud rendering. At the same time, the ability to hallucinate anatomically plausible content over large unobserved regions can be beneficial for data augmentation, as it increases viewpoint and appearance diversity without additional scanning effort. Second, our pipeline assumes minimal tracking drift. Larger pose errors introduce geometric distortions into the reconstructed volumes. Although small tracking inaccuracies may be visually tolerable, that is, overlap with normal anatomical variability, it still leads to erroneous quantitative measurements. However, implementing 3D operations within a differentiable framework opens the door to minimizing the reprojection error for pose refinement and improved geometric consistency, which we plan to explore in future work.

In summary, the proposed pipeline achieves high-quality, real-time novel-view ultrasound synthesis by integrating point-cloud rendering with efficient diffusion-bridge completion. The framework offers a principled way to enrich training datasets for downstream tasks such as segmentation, registration, and quality assessment, and also supports interactive, view-adaptive ultrasound visualization.

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
