# OpenReview forum: "Real-Time Novel-View Freehand Ultrasound Imaging via Point-Cloud Rendering and Diffusion-Bridge Completion"
_MIDL.io/2026/Conference — MIDL 2026 Poster_

### Official Review · Reviewer_wGjN · 2026-01-09

**Confidence:** 4
**Preliminary Rating:** 3
**Final Rating:** 3

**Summary:**

This paper proposes a real-time novel-view freehand ultrasound imaging framework that combines geometry-aware point-cloud rendering with Schrödinger Bridge–based diffusion completion. Partial novel views rendered from sparse freehand acquisitions are completed using an Image-to-Image Schrödinger Bridge (I2SB) model, including a latent-space variant that enables efficient high-resolution inference. Experiments on a large abdominal ultrasound dataset demonstrate improved perceptual quality and inference speed compared to a diffusion-based inpainting baseline.

**Strengths:**

The paper addresses an interesting application setting by using generative modeling to refine imperfect novel views rather than synthesizing them from scratch. The use of an I2SB model to focus generation on missing regions, while preserving observed pixels, is a sensible design choice and well suited for medical reconstruction tasks that require pixel-level accuracy.

**Weaknesses:**

1) Pipeline description and completeness. The main pipeline figure and description do not fully convey the proposed method. Important intermediate representations and processing steps are not clearly illustrated. It would be helpful to include a more detailed step-by-step breakdown of the pipeline, potentially with intermediate outputs and algorithmic details provided in the appendix.

2) Baseline selection and fairness. The baseline comparison is limited and raises concerns about fairness. While the authors claim superior reconstruction quality and inference speed over “standard diffusion-based completion,” RePaint is primarily an inference-time diffusion method and may not represent a standard conditional diffusion baseline. To more convincingly demonstrate the advantage of diffusion-bridge conditioning, comparisons with conditional diffusion models, ControlNet-style approaches, or other relevant baselines would be more appropriate. It is also unclear why traditional similarity-based methods (e.g., NCC-based interpolation or completion) are not included.

3) Latent I2SB implementation details. The experimental description of the latent I2SB variant lacks sufficient detail. In particular, it is unclear what data were used to pretrain the VAE and whether this pretraining overlaps with the evaluation data. Providing clearer information on VAE training and integration would improve reproducibility and interpretation.

4) Qualitative results. In Figure 4, the visual differences between the three compared methods are not clearly distinguishable. Stronger qualitative evidence or more challenging examples would help support the claimed advantages of the proposed approach.

5) Uncertainty estimation analysis. The uncertainty estimation experiments are not clearly described. It is unclear whether the reported uncertainty results are based on a single example or aggregated across the entire test set or a specific subgroup. Additionally, it is ambiguous whether uncertainty is measured relative to ground truth (where available) or solely as variance across generated samples. Clarifying the experimental protocol and interpretation of uncertainty would strengthen this part of the analysis.

6) Definition of real-time performance. While the authors claim that the proposed method operates in “real time,” it is unclear whether the reported timing measurements include the entire pipeline (e.g., point-cloud rendering, voxelization, and slice extraction) or only the diffusion-based completion stage. Clarifying what components are included in the runtime evaluation, and reporting end-to-end latency, would provide a more accurate assessment of real-time applicability.

**Detailed Comments:**

1) Clarity and writing quality. The paper would benefit from substantial improvements in writing clarity and presentation. Several key methodological details are insufficiently explained, making parts of the paper difficult to follow. There are also multiple grammatical issues and undefined notations; for example, the quantities  𝐹 and  𝐷 introduced on page 4 are not clearly defined. Improving clarity, consistency, and notation would significantly enhance readability.

2) Related work coverage. The paper cites a relatively small number of references (14 in total), and the discussion of related work, particularly diffusion-based methods applied to ultrasound, is limited and somewhat outdated. In addition, prior work on ultrasound inpainting using GAN-based approaches is largely omitted. A more comprehensive and up-to-date review would better situate the contribution within the existing literature.

3) It may be preferable to highlight the best-performing results using symbols other than an asterisk, as asterisks are typically reserved to denote statistical significance.

**Justification Of Final Rating:**

The authors respond to the points raised in my review with clarifications and acknowledgments of the limitations of the current submission. While these responses help clarify the authors’ perspective, the key concerns regarding writing quality, technical accuracy, baseline selection, and experimental justification remain insufficiently addressed in the current manuscript. My overall assessment therefore remains unchanged.

**Justification Of The Preliminary Rating:**

From the perspective of application relevance and methodological choice, the reviewer finds the work appealing. The major concerns primarily lie in the clarity of presentation and the experimental details rather than the core idea itself. Therefore, the preliminary rating is borderline, with the potential to be raised depending on how well these issues are addressed in the rebuttal.

**Questions To Address In The Rebuttal:**

See weeknesses and detailed comments

---

> ### Author Response · Authors · 2026-01-24
>
> Thanks for the constructive feedback and detailed observations. We appreciate the suggestions that help clarify several aspects of the paper. We respond to these points below.
>
> **Confusion about Pipeline Description and Completeness**
>
> We apologize for the confusion caused by the earlier figure. We have revised the pipeline diagram to more clearly depict the workflow and have added a detailed step-by-step breakdown of the method in Figure 2, including inputs and outputs.
>
> **Baseline Selections**
>
> Our initial choice of RePaint was primarily intended to illustrate that standard diffusion-based inference, whether conditional or non-conditional, can be computationally expensive. Regarding reconstruction performance, we have now added experiments to compare diffusion-bridge conditioning against a conditional diffusion model (Palette) as well as a traditional interpolation method. These results have been incorporated into the revised manuscript.
>
> **Latent I2SB Implementation Details**
>
> For the latent I2SB experiments, we use a standard pretrained VAE used in Stable Diffusion, which is trained on natural images. We do not perform any additional fine-tuning, and therefore the VAE pretraining is completely independent of both our training and evaluation datasets. We have clarified these details in the revised manuscript to improve reproducibility.
>
> **Qualitative Results**
>
> We agree that the visual differences between the baseline and the bridge method are relatively subtle and require close inspection. This is expected, as diffusion-based models are already very powerful, and for small inpainting regions even the baseline approach can produce visually plausible results. To better support the advantages of the bridge formulation, we therefore report multiple quantitative metrics as well as uncertainty maps, which demonstrate that the bridge method yields higher-quality and more stable reconstructions. In addition, we show that the bridge method achieves comparable or slightly improved visual performance in a shorter inference time compared to the baseline.
>
> **Uncertainty Estimation Analysis**
>
> The reported uncertainty maps are computed per example, and are not defined relative to ground truth.
> The presented uncertainty is the posterior variance learnt by the model. It is computed by Monte Carlo sampling: for one given, perform inpainting multiple times and compute the mean / variance of the output across all samples. It is primarily used to validate that the model has not learnt a deterministic mapping in cases where the input information should be too sparse to fully determine the output. We clarified the experimental protocol and interpretation in the revised manuscript.
>
>
> **Definition of real-time performance**
>
> The reported runtime of <0.2 s refers specifically to novel-view generation during inference, which consists of reslicing followed by diffusion-based inpainting. The diffusion model dominates this cost, reslicing alone takes <10 ms.
>
> Initial volume construction (point-cloud fusion and voxelization) is accounted for separately because it is a one-time preprocessing step that can be reused to generate many different views from the same sweep. We have revised Fig. 2 to make this separation explicit. In practice, volume generation takes ~1 ms per 2D slice. Those timings have been added to the revised manuscript.
>
>
> **Other Detail Concerns**
> 1. Clarity and writing quality.  We have revised the manuscript to improve clarity, consistency, and overall presentation. In particular, we expanded key methodological descriptions, defined missing notations (including F and D on page 4), and corrected grammatical issues. We believe these changes significantly enhance readability.
> 2. Related work coverage. We have expanded the related work section to include a more comprehensive and up-to-date review of diffusion-based ultrasound methods as well as prior GAN-based ultrasound inpainting approaches.
> 3. Highlight Symbols. We have updated the table and now highlight the best-performing results using symbols other than asterisks.

---

> > ### Comment · Reviewer_wGjN · 2026-02-01
> >
> > Thank you for the authors’ response. However, several of my concerns remain unresolved. Unfortunately, both the revised manuscript and the rebuttal continue to exhibit a lack of care in writing, technical accuracy, and experimental justification, which raises serious concerns about the rigor with which the work has been prepared and revised.
> >
> > **Writing Quality**:
> >
> > As an example highlighted by the authors in the revised manuscript:
> >
> > “Palette is better then RePaint due to the beneficial of conditional training.”
> >
> > This sentence contains multiple grammatical errors and unclear phrasing. Clear writing, correct grammar, and the absence of typographical errors should be basic requirements for an MIDL submission. I initially treated the poor writing quality in the first version as a consequence of limited revision time and expected improvement in the revised manuscript. However, the persistence of such issues in both the revision and the rebuttal suggests that the manuscript and the review process have not been treated with sufficient care. This significantly undermines confidence in the overall rigor of the work.
> >
> > **Baseline Selection and Technical Accuracy**
> >
> > I do not understand what is meant by “diffusion-bridge conditioning(Palette)” in the rebuttal. To the best of my knowledge, Palette is not a diffusion-bridge–based model. This appears to be either a technical misunderstanding or an imprecise statement. In either case, it raises concerns about the authors’ technical expertise and carefulness in describing both the baseline methods and their own approach.
> >
> > Moreover:
> >
> > - Palette is not properly cited or discussed in the revised manuscript.
> >
> > - As a major baseline, it lacks a clear methodological explanation.
> >
> > - Given that it is now 2026 and many more advanced conditional diffusion models have been proposed since 2021, using Palette as the only conditional diffusion–based baseline is not convincing.
> >
> > **Comparison to Traditional Baselines**
> >
> > The traditional baseline methods demonstrate **comparable qualitative and quantitative performance** to the proposed approach while achieving much faster inference speed. Under these circumstances, the practical necessity of the proposed method is unclear.
> >
> > The revised manuscript claims:
> >
> > “Linear interpolation does not involve model inference and is therefore the fastest method, but it inevitably loses realistic fine-scale details. .”
> >
> > This claim is not persuasive without clear qualitative evidence. In fact, the qualitative results shown are extremely similar between diffusion-based methods and traditional baselines. Additionally, the degradation observed in the task appears to be largely linear or small-scale, rather than involving large missing regions, which further weakens the argument that diffusion-based modeling provides a clear advantage in this scenario.
> >
> > **Qualitative Results and Task Difficulty**
> >
> > In the rebuttal, the authors attribute the similarity in qualitative performance to the power of diffusion-based models. However, given the strong qualitative and quantitative performance of simple linear interpolation, the reviewer instead believes that this task represents a relatively simple application scenario. As a result, the practical necessity and added value of the proposed method remain questionable.
> >
> > **Uncertainty Estimation Analysis**
> >
> > Several claims regarding uncertainty estimation are insufficiently supported:
> >
> > - The statement “Both pixel-space and latent I2SB models produce significantly lower and more localized variance” is not backed by any rigorous or statistical analysis in the revised manuscript.
> >
> > - Given that Palette exhibits higher variance, it is unclear whether this translates into better performance under any realistic scenario.
> >
> > - In the “Step Reduction for Real-Time Completion” section, RePaint—already shown to perform worst in both reconstruction quality and inference speed—is used as the comparison baseline. This comparison is unfair; a more meaningful comparison should be made against Palette instead.
> >
> > **Related Work Coverage**
> >
> > The revised manuscript still does not include a dedicated Related Work section. Furthermore, only three papers published after 2024 are cited. This falls far short of a comprehensive or up-to-date review of the relevant literature, especially given the rapid development of diffusion-based image-to-image models in recent years.

---

> > > ### Author Response · Authors · 2026-02-01
> > >
> > > We thank the reviewer for the detailed feedback and take these concerns seriously. Below, we clarify points that may have caused confusion.
> > >
> > > **Writing Concerns**
> > >
> > > We thank the reviewer for pointing out the writing issues. We agree that the example sentence contains multiple grammatical errors and should not have appeared in either the revised manuscript or the rebuttal. While we did make language revisions in earlier versions, we acknowledge that these corrections were not sufficiently thorough, and that remaining issues should have been identified and resolved before resubmission. We regret underestimating the negative impact that such errors can have on the perceived rigor of the work.
> > > We have therefore conducted a more stringent and systematic language review across the entire manuscript, correcting grammar, wording, and typographical issues, and will ensure professional proofreading before the camera-ready version.
> > >
> > >
> > > **Clarification for Palette**
> > >
> > > We would like to clarify the source of the confusion regarding the phrasing “diffusion-bridge conditioning (Palette)” in the rebuttal. Our intention was to compare diffusion-bridge–based conditioning (I2SB) against conditional diffusion models, with Palette as a representative example.
> > > However, placing “Palette” in parentheses immediately after “diffusion-bridge conditioning” was misleading and incorrectly suggested that Palette belongs to the diffusion-bridge category.
> > > We acknowledge that this was an imprecise use of parentheses and terminology on our side, and we apologize for the confusion caused. For clarity, in the revised manuscript, Palette is consistently treated and cited as a conditional diffusion–based inpainting method, rather than a diffusion-bridge approach.
> > >
> > > **Comparison to Traditional Baseline**
> > >
> > > We acknowledge that in the current experimental setting, the missing regions are relatively moderate, and simple interpolation baselines already perform reasonably well in both qualitative and quantitative metrics. At the same time, we would like to highlight a fundamental limitation of traditional methods: their completions are deterministic and tend to become overly smooth as the extent of missing regions increases.
> > > In contrast, diffusion-bridge models enable stochastic and diverse completions, which can be particularly valuable for data augmentation and downstream learning tasks, where exposure to multiple plausible anatomical realizations may improve robustness and generalization.
> > >
> > > **Uncertainty Analysis**
> > >
> > > Our uncertainty analysis is intended as a qualitative characterization, rather than a formal statistical comparison. We agree that a rigorous quantitative evaluation of uncertainty would require additional downstream tasks or statistical tests, which are beyond the scope of the current submission.
> > >
> > >
> > > **Related Work Coverage**
> > >
> > > We agree that the related work coverage could be further strengthened. Due to the scope and stage of the current submission, we treat this as a limitation of the present manuscript and will take this feedback into careful consideration in future revisions.

---

### Official Review · Reviewer_5ZPQ · 2026-01-10

**Confidence:** 4
**Preliminary Rating:** 4
**Final Rating:** 4

**Summary:**

1. This paper proposes a real-time pipeline for novel-view freehand ultrasound synthesis by combining point-cloud–based geometric rendering with diffusion-bridge image completion.
2. Acquired B-mode slices with tracked probe poses are projected into a 3D point cloud, from which novel views are rendered as partially observed slices with visibility masks derived from voxel density.
3. An Image-to-Image Schrödinger Bridge (I2SB) model completes missing regions while conditioning on visible anatomy, enabling fast and stable synthesis. A latent I2SB variant further reduces computation, achieving high-resolution inference in under 0.2 seconds per view.

*Experiments on a large abdominal ultrasound dataset show improved visual fidelity, perceptual metrics, uncertainty stability, and runtime compared to diffusion inpainting baselines.*

**Strengths:**

1. *Clear & Relevant Problem Setting*: The paper addresses sparse sampling and limited probe coverage in freehand ultrasound, which is a well-known practical limitation.

2. *Well-designed Hybrid pipeline*: The separation between geometry (point-cloud rendering) and appearance (diffusion-bridge completion) is conceptually sound and effective.

3. *Appropriate choice of diffusion Bridge*: Conditioning on partially observed slices rather than noise leads to faster convergence, improved stability, and better preservation of observed anatomy.

4.  *Real-time Performance* : Step-reduction experiments show saturation at ~5 diffusion steps, enabling interactive use.

5.  *Latent-space acceleration*: This is practical and the latent I2SB variant preserves anatomical structure while significantly reducing computation.

6. *Comprehensive Evaluation*: Includes perceptual metrics, uncertainty analysis, and runtime comparisons, providing a balanced assessment.

**Weaknesses:**

1. All experiments rely on a proprietary abdominal ultrasound dataset, preventing independent verification.
2. The impact on navigation, guidance, or downstream clinical tasks is not assessed.
3.  Point-cloud reconstruction and diffusion-based completion are established techniques; novelty lies mainly in their integration and optimization.
4. When large regions are unobserved, the diffusion model may generate multiple plausible but uncertain completions, which could be problematic in safety-critical settings.

**Detailed Comments:**

1. The density-thresholding strategy for visibility mask generation is practical. Please consider reporting sensitivity to this threshold choice would further strengthen robustness claims.
2. Memory usage and GPU requirements could be reported alongside inference time to better characterize deployment feasibility.
3. A brief discussion on how the method might behave under probe tracking errors or drift will increase the quality of the paper.
4. Definitely fix minor editorial issues (typos and repeated section numbering) could be corrected for clarity.

**Justification Of Final Rating:**

Thank you for the detailed and thoughtful rebuttal.

1. The authors have satisfactorily addressed my original questions regarding robustness to tracking errors, behavior in largely unobserved regions, sensitivity to density thresholding, and the interpretation of uncertainty.
   * I appreciate the clear and responsible discussion acknowledging the limitations of diffusion-based completion in safety-critical settings, as well as the clarification that the proposed method neither corrects nor amplifies geometric errors introduced by tracking drift.

2. The additional clarifications on runtime accounting, evaluation metrics, baseline selection, and latent $\mathrm{I}^{2}\mathrm{SB}$ implementation also address concerns raised by other reviewers regarding pipeline completeness, fairness of comparison, and reproducibility.

**Overall, the rebuttal strengthens the manuscript and reinforces my original assessment.**.
*My overall evaluation remains unchanged.*

**Justification Of The Preliminary Rating:**

1. This paper presents a well-engineered and practically relevant solution for real-time novel-view ultrasound synthesis from sparse freehand acquisitions.
2. The integration of point-cloud rendering with diffusion-bridge completion is thoughtfully designed and convincingly evaluated, particularly in terms of speed–quality trade-offs.
3. While the work is largely incremental at the component level and relies on private data without downstream clinical validation, the overall system demonstrates clear utility and technical maturity.
4. The paper should be of interest to researchers working on ultrasound reconstruction and real-time medical image synthesis.

**Questions To Address In The Rebuttal:**

1. How robust is the pipeline to probe pose noise or tracking drift, and could diffusion-bridge completion amplify such errors?
2. Do you  anticipate challenges when applying the method to cardiac or musculoskeletal imaging?
3. How might clinicians interpret or utilize the uncertainty estimates, especially when large regions are unobserved?

---

> ### Author Response · Authors · 2026-01-24
>
> Thanks for the encouraging comments and thoughtful suggestions! We address the raised concerns in the following responses.
>
> **Uncertainty in Largely Unobserved Regions**
>
> We agree that hallucination tends to occur in largely unobserved regions. For practical use, the missing rate needs to be constrained, which can be achieved by increasing the sweep coverage and fusing multiple sweeps. For downstream tasks, this uncertainty can be leveraged for data augmentation, i.e., by sampling diverse novel views.
>
> **Sensitivity to Threshold Choice**
>
> The need for a threshold is linked to pytorch3d integration. PyTorch3D uses its own interpolation kernel during voxelization (not exposed to users to our knowledge) and attempts to fill empty voxels by linear interpolation quite far from the input points. Density is computed by summing interpolation weights rather than integer point counts so those interpolated voxels have a low density. Setting a non-zero density threshold masks those back to 0 and lets the AI fill them rather than linear interpolation. We agree that automating the threshold selection would make it easier to generalize this pipeline to different imaging settings with varying input/output resolutions and frame rates.
>
> **Robustness to Tracking Errors**
>
> This is an important and practical question. Tracking errors manifest as geometrical distortions in the reconstructed volume. The presented inpainting method does not change that volume so it will neither amplify nor correct such distortions. Small tracking errors may overlap with anatomical variability. In that regime, inpainting can still generate realistic images but measurements computed on them would be incorrect.
>
> While we do not expect inpainting to increase the magnitude of errors, it could make them harder to spot if one cannot visually tell acquired slices apart from through-plane reconstructions. This should be addressed at the application level by separate labeling and/or routing of native slices and reconstructions to ensure measurements are only computed on native images.
>
> Larger tracking will generate artifacts in the initial volume that the current method will not correct either. That is why the current results focus on reconstructions from single sweeps rather than longer exams with higher drift risks, and why we chose to base our approach on a differentiable point cloud rendering engine to prepare the way for geometric correction in future work. Those points have been added to the discussion.
>
>
> **Generalization to Cardiac / MSK**
>
> While the inpainting method is likely to generalize if trained on appropriate data, application to cardiac imaging may require adding cardiac motion to our current static volume reconstruction. Building that model would require tracking the cardiac phase in the input scan and binning slices from different heart beats to collect sufficient spatial coverage. Fusing slices acquired at longer times apart requires higher accuracy from the tracking system to keep geometric errors under control.
> Musculoskeletal imaging, on the other hand, is characterized by anisotropic and directionally aligned muscle fibers, as well as a mix of bone, tendon, and soft tissue with heterogeneous contrast. These properties may increase the difficulty of maintaining geometric and textural consistency. However, with sufficient training data capturing these structural patterns, we expect the model to generate realistic and anatomically plausible completions.
> In both cases, imaging around bone may increase the risk of shading artifacts. The current max compounding of slices partially alleviates shading by allowing a better slice to fill in shaded areas. Fully correcting shading may not be necessary for our main goal of generating training data: if shading is going to happen at test time, then it should be in the training data as well.
>
> **Intepretability of Uncertainty Maps**
>
> This is a hard problem that encompasses both engineering (is variance a good measure of image reliability?) and human-computer interaction (how to present the information to sonographers so that it is helpful rather than distracting?), and to our knowledge that second step has so far precluded clinical application of uncertainty maps. For practical clinical use, it may be more straightforward to use Bayesian programming to assess the posterior variability of clinically relevant features (i.e., pass multiple generated completions through a task model and compute statistics on the task outputs rather than image pixels). This is an important but orthogonal direction, and we view it as future work beyond the scope of the present submission.
>
> **Other Detail Concerns**
>
> Memory usage and GPU requirements have been reported in the revised manuscript.
> Typos and section numbering have been corrected.

---

> > ### Comment · Reviewer_5ZPQ · 2026-01-24
> > **Response to Rebuttal and Reviewer Discussion**
> >
> > Thank you for the detailed and thoughtful rebuttal.
> >
> > 1. The authors have satisfactorily addressed my original questions regarding robustness to tracking errors, behavior in largely unobserved regions, sensitivity to density thresholding, and the interpretation of uncertainty.
> >    * I appreciate the clear and responsible discussion acknowledging the limitations of diffusion-based completion in safety-critical settings, as well as the clarification that the proposed method neither corrects nor amplifies geometric errors introduced by tracking drift.
> >
> > 2. The additional clarifications on runtime accounting, evaluation metrics, baseline selection, and latent $\mathrm{I}^{2}\mathrm{SB}$ implementation also address concerns raised by other reviewers regarding pipeline completeness, fairness of comparison, and reproducibility.
> >
> > **Overall, the rebuttal strengthens the manuscript and reinforces my original assessment.**.
> > *My overall evaluation remains unchanged.*

---

### Official Review · Reviewer_kBYy · 2026-01-15

**Confidence:** 4
**Preliminary Rating:** 3
**Final Rating:** 4

**Summary:**

The paper proposes a real-time pipeline for novel-view synthesis in freehand ultrasound by combining geometry-aware point-cloud rendering with a conditional diffusion-bridge completion model (I²SB). The approach renders partially observed slices from a reconstructed point cloud and completes them via a Schrödinger-bridge-based inpainting model, with a latent variant to enable high-resolution, low-latency inference. On a multi-subject abdominal dataset, the method reportedly achieves realistic visuals, improved structural continuity, and <0.2 s per view, outperforming a diffusion inpainting baseline (RePaint) in both speed and fidelity.

**Strengths:**

1. Integrates a pragmatic, geometry-aware point-cloud reslicing pipeline with a conditional diffusion bridge tailored to ultrasound inpainting, which is a sensible and novel pairing for this domain.
2. The use of a latent I²SB variant to achieve full-resolution real-time performance is well motivated and practically relevant.
3. Conditioning the diffusion process on visibility masks (instead of starting from noise) is a reasonable application of Schrödinger bridge ideas for efficient, constrained synthesis in medical imaging.
4. Real-time, view-adaptive ultrasound visualization is of substantial clinical interest for navigation, improved coverage, data augmentation, and robustness to sparse sweeps.

**Weaknesses:**

1. For true novel-view synthesis, evaluation relies on perceptual metrics (FID/LPIPS) that may not correlate with anatomical correctness in ultrasound. The grounding truth seems to be a bit ambigious.
2. While diffusion bridges are conditioned on visible regions, the method can still hallucinate anatomy in large missing areas, which is risky in medical settings.
3. The exact runtime accounting is ambiguous: timing seems to cover the denoising steps, but it is unclear whether point-cloud rendering and reslicing are included in the <0.2 s claim.

**Detailed Comments:**

1. The paper shows convincing results visually, and the evaluation of true novel-view synthesis relies primarily on perceptual metrics such as FID and LPIPS. However, no ground-truth novel views exists. It seems a bit unclear how well these metrics correlate with anatomical correctness in ultrasound imaging. Further clarification of how these metrics are computed in this setting, and discussion of their limitations would strengthen the credibility of the reported improvements.
2. The paper emphasizes real-time performance with a reported inference time of <0.2 s per view, which is an attractive property. However, it is currently unclear whether this timing includes the full pipeline.

**Justification Of Final Rating:**

After reading the rebuttal, I am willing to give a weak accept. The clarification on the evaluation protocol makes the use of LPIPS and FID better contextualized, and although these metrics remain imperfect for assessing anatomical correctness, their role as perceptual measures is now clearer. The authors’ explanation that the method is intended more for generative priors and data augmentation than direct clinical use also helps frame the hallucination concern, even if the underlying risk remains. The runtime breakdown significantly improves the credibility of the real-time claim by clarifying what is included in the <0.2 s figure. Overall, the rebuttal resolves the main ambiguities that limited my confidence, and I therefore support a weak accept.

**Justification Of The Preliminary Rating:**

I assign a borderline rating because this paper presents a sensible and practically motivated approach, combining geometry-aware point-cloud reslicing with a conditional diffusion bridge for novel-view ultrasound synthesis. The latent I²SB variant and the use of visibility-conditioned diffusion are well motivated and support the goal of real-time, view-adaptive visualization, which is of clear clinical interest. However, the evaluation for true novel views relies primarily on perceptual metrics such as FID and LPIPS, whose ability to reflect anatomical correctness in ultrasound remains unclear. In addition, although conditioning on visible regions constrains the generation process, the risk of anatomically plausible but incorrect hallucinations in large unobserved areas is not sufficiently addressed. Finally, the reported <0.2 s runtime lacks clarity regarding whether the full pipeline, including point-cloud rendering and reslicing, is included. Overall, the work shows promise, but these unresolved issues limit confidence in its reliability and practical impact.

**Questions To Address In The Rebuttal:**

1. How exactly are LPIPS and FID computed for novel views without ground truth? How do you avoid or reduce potential hallucinations?
2. Does the <0.2 s per view runtime include point-cloud voxelization, density mask computation, and reslicing, or only the diffusion-bridge completion?

---

> ### Author Response · Authors · 2026-01-24
>
> Thanks for the practical recognition of our pipeline and for raising the confusion! We address the questions and suggestions in details below.
>
> **Evaluation in FID/LPIPS**
>
> LPIPS is used for supervised perceptual evaluation on synthetically masked images (i.e., ground truth images multiplied by a density mask computed on a different pose). FID is used for unsupervised perceptual evaluation on real through-plane reconstructed slices for which the ground truth is unknown. We have split Table 1 into supervised and unsupervised metrics for clarification. While both metrics rely on downstream networks trained for natural images rather than medical images, they offer the advantage of standardization and reproducibility.
>
> FID measures whether two sets of images differ in perceptual appearance. As an unsupervised perceptual metric, FID takes structural/anatomical plausibility into account, i.e., whether shapes inpainted in a given image appear similar to those in some (other) images of the ground truth dataset. Verification of anatomical correctness, however, would require downstream task-based evaluation, such as segmentation, landmark tracking, or quantitative physiology estimation, which are outside the scope of this paper and typically rely on domain-specific clinical datasets and expert annotations.
>
>
>
> **Hallucinations in Large Missing Area**
>
> We agree that hallucinations are not avoidable when inpainting large missing areas. Rather than using inpainted images directly for clinical decisions, one of the main aims of our work is to use inpainting to generate training data for downstream tasks. In that context, stochastic hallucinations can be treated as a form of data augmentation as long as one can generate enough variability. That is why the speed of the pipeline matters: our goal is to make it fast enough to generate new images online during training of downstream tasks, thus preventing overfitting to hallucinations since the training will experience new completions at every epoch.
>
> **Runtime Clarification**
>
> The reported runtime of <0.2 s refers to novel view generation, i.e. reslicing and inpainting inference. That time is dominated by the AI model, reslicing takes less than 10 milliseconds.
>
> Initial volume construction is counted separately because it only needs to be computed once as a pre-processing, then can be re-used to generate multiple different views from a given volume. We have updated the pipeline illustration (Fig. 2) to more clearly convey this distinction. Volume generation takes about 1ms per slice, e.g. 0.5s to fuse 500 slices into one volume. We have added those measurements to the paper.

---

> > ### Comment · Reviewer_kBYy · 2026-01-28
> >
> > After reading the rebuttal, I am willing to raise my score from borderline to weak accept. The clarification on the evaluation protocol makes the use of LPIPS and FID better contextualized, and although these metrics remain imperfect for assessing anatomical correctness, their role as perceptual measures is now clearer. The authors’ explanation that the method is intended more for generative priors and data augmentation than direct clinical use also helps frame the hallucination concern, even if the underlying risk remains. The runtime breakdown significantly improves the credibility of the real-time claim by clarifying what is included in the <0.2 s figure. Overall, the rebuttal resolves the main ambiguities that limited my confidence, and I therefore support a weak accept.

---

### Author Rebuttal · Authors · 2026-01-25

**Rebuttal:**

Across Reviewers, the main concerns focus on two aspects of the submission:

**1. Clarity of Pipeline Description & Runtime Accounting.**
Reviewers noted confusion regarding the exact workflow and runtime definition. We addressed this by revising the pipeline figure, adding step-by-step illustrations, and explicitly separating one-time volume construction from per-view novel-view generation. Precise timing numbers were also incorporated into the manuscript.

**2. Uncertainty and Hallucinations in Large Missing Areas.**
We clarified how uncertainty is computed and interpreted, and discussed the intrinsic hallucination behaviors that arise in largely unobserved regions. We also clarified in the manuscript that our primary intention is to enable training data augmentation and to support downstream tasks, rather than to use the hallucinated completions directly for clinical interpretation.


Beyond these common concerns, in response to Reviewer 5ZPQ’s comments on robustness to tracking drift, we provide a more detailed analysis of its potential impact in our discussion. In response to Reviewer wGjN’s comments regarding related work augmentation and baseline selection, we expanded the literature review and added comparisons against both a traditional interpolation-based method and a conditional diffusion model (Palette), enabling a more complete evaluation of reconstruction performance and inference efficiency.

Additionally, we performed another proofreading pass to improve grammar, correct typographical errors, and refine wording for overall clarity and readability. Major textual changes in the revised manuscript have been highlighted in yellow, and modified figures have been marked with red bounding boxes for visibility.

**Supporting Material:**

/attachment/ef92e08b2b5242e0d5f122c2e8657878dcf75c31.pdf

---

### Comment · Area_Chair_8iQj · 2026-01-25
**the paper open for discussions**

Dear Reviewers,

The authors have submitted their responses to the comments you raised. The paper is now open for discussion.

---

### Meta-Review · Area_Chair_8iQj · 2026-02-06

**Recommendation:** Accept (Poster)
**Confidence:** 5

**Metareview:**

I believe the authors have addressed most of the major concerns raised by the reviewers. Two reviewers have updated their scores to weak accept, while one reviewer remains borderline, primarily due to remaining writing and clarity issues. Overall, the contribution is important and will likely generate valuable discussion during the meeting. I encourage the authors to carefully revise the final camera‑ready version, including a thorough grammar check and full read‑through.

---

### Decision · Program_Chairs · 2026-02-13

Accept (Poster)